Additional records and stratigraphic distribution of the middle Eocene carettochelyid turtle Anosteira pulchra from the Uinta Formation of Utah, North America

Adrian Brent badria@midwestern.edu 1
Holroyd Patricia A. 2
Hutchison J. Howard 2
Townsend KE Beth 1
1 Department of Anatomy, Midwestern University , Glendale , AZ , United States of America
2 Museum of Paleontology, University of California, Berkeley , Berkeley , CA , United States of America
Knoll Fabien
Electronic publication date: 2020 Aug 24
Publication date: 2020
Volume: 8
Electronic Location ID: e9775
Received 2020 Jun 15; Accepted 2020 Jul 30
Copyright: ©2020 Adrian et al.
Copyright year: 2020
Copyright holder: Adrian et al.
License: This is an open access article distributed under the terms of the Creative Commons Attribution License, which permits unrestricted use, distribution, reproduction and adaptation in any medium and for any purpose provided that it is properly attributed. For attribution, the original author(s), title, publication source (PeerJ) and either DOI or URL of the article must be cited.
License URL: https://creativecommons.org/licenses/by/4.0/

Keywords: Uinta formation, Turtle, Biostratigraphy, Anosteira pulchra, Carettochelyidae

Funding: Midwestern University faculty intramural funds Funding for this research was provided by Midwestern University faculty intramural funds to K.E. Beth Townsend. The funders had no role in study design, data collection and analysis, decision to publish, or preparation of the manuscript.

==============================
Background

Anosteira pulchra is one of two species of the obligately-aquatic freshwater clade Carettochelyidae (pig-nosed turtles) from the Eocene of North America. Anosteira pulchra is typically rare in collections, and their distribution is poorly documented. The Uinta Formation [Fm.] contains a diverse assemblage of turtles from the Uintan North American Land Mammal Age. Whereas turtles are abundantly preserved in the Uinta Fm., A. pulchra has been reported only from a few specimens in the Uinta C Member.

Methods

We describe new records of Anosteira pulchra from the Uinta Basin and analyze the distribution of 95 specimens from multiple repositories in the previously published stratigraphic framework of the middle and upper Uinta Fm.

Results

Here we report the first records of the species from the Uinta B interval, document it from multiple levels within the stratigraphic section and examine its uncommon appearance in only approximately 5% of localities where turtles have been systematically collected. This study details and extends the range of A. pulchra in the Uinta Fm. and demonstrates the presence of the taxon in significantly lower stratigraphic layers. These newly described fossils include previously unknown elements and associated trace fossils, with new anatomical information presented. This study provides insight into the taxonomy of Anosteira spp. in the middle Eocene, and suggests the presence of a single species, though no synonymy is defined here due to limits in Bridger material.

Figure 1 Index map of Utah and collection sites of Anosteira pulchra in the current study.

Introduction

The Uinta Formation [Fm.] in the Uinta Basin of northeastern Utah (Fig. 1) contains a rich and diverse assemblage of turtles from the late middle Eocene Uintan North American Land Mammal Age (NALMA). Anosteira is a genus of small to medium-sized highly aquatic freshwater turtles belonging to Carettochelyidae (Gill, 1889) that apparently emigrated from Asia to North America during the early Bridgerian NALMA (Hutchison, 1998). Two North American species of the genus have been described to date. The older of the two, Anosteira ornata, is known from several Bridgerian sites in southwest Wyoming (see Joyce, 2014 for a recent summary). Gilmore (1916) provisionally reported A. ornata in Uinta C based on CM 2954, collected on the White River near Ouray, Utah. Clark (1932) named Pseudanosteira pulchra based on CM 11808 from the Uinta C horizon at Leota Ranch, northwest of Ouray, Utah, but did not mention CM 2954. Broin (1977) recombined P. pulchra as A. pulchra, noting the differentiation of Pseudanosteira from Anosteira on the shape of the anterior neurals, but reduction of the vertebral scales was not supportable in the absence of data on individual and specific variability. This synonymy was followed by Joyce (2014) and Joyce, Volpato & Rollot (2018), and is followed here. Joyce (2014) noted the potential range extension represented by CM 2954 but did not elect to make a species assessment. As the literature currently stands, only two carettochelyid specimens have been noted or described from the Uinta Basin. Both occur in the upper part of the Uinta Fm., in beds historically referred to Horizon C or Uinta C, and may represent two different species. However, targeted collecting in recent years of Uintan herpetofauna in a measured stratigraphic framework has yielded 95 carettochelyid specimens, none of which have previously been described. The aim of this study is to describe the stratigraphic and geographic distribution of A. pulchra in the Uinta Fm. and provide new anatomical information on its morphology.

Geological Setting

The Uinta Basin in northeastern Utah (Fig. 1) is approximately 135 miles wide along its east–west axis and 100 miles across from north to south, encompassing an area of 10, 943 km2 (Ryder, Fouch & Elison, 1976; Prothero, 1996; Murphey et al., 2011). Its boundaries include the Uinta Mountains to the north, the Book Cliffs/Tavaputs Plateau to the south, the Douglas Creek Arch and Roan Plateau to the east, and the Wasatch Range to the west (Murphey et al., 2011) (Fig. 1). Over 4,500 m of Eocene sediments accumulated during the Laramide orogenesis, filling the Uinta, Green River, and Piceance Creek basins (Prothero, 1996; Murphey et al., 2011). These sediments record part of a vast system of middle Eocene lakes that covered a large portion of northeastern Utah, southwestern Wyoming, and western Colorado (Ryder, Fouch & Elison, 1976; Prothero, 1996; Murphey et al., 2011; Chamberlain et al., 2012).

During the Bridgerian NALMA (47–49 Ma), the Green River lake system began to recede, replacing lacustrine shales with fluvial-deltaic mudstones and sandstones which now comprise a rich matrix for terrestrial fossil vertebrates (Murphey et al., 2011). In the Uinta Basin, the fluvial Uinta Fm. gradually replaced the Green River lake system, beginning at the east end of the basin (Fig. 1). As a result, the lower fluvial sandstones of the eastern Uinta Fm. are laterally equivalent to lacustrine evaporates, sandstones, and limestones in the western Uinta Basin, and the two units share complex interfingering (Dane, 1954; Dane, 1955; Ray, Kent & Dane, 1956; Cashion, 1967; Ryder, Fouch & Elison, 1976). The primary focus of this study is to describe the stratigraphic distribution of Anosteira pulchra in the eastern Uinta Fm., but we also record some additional western occurrences (Fig. 1).

Figure 2 Stratigraphic distribution of A. pulchra in the upper Uinta Fm.

(A) Stratigraphic sections indicating marker unit correlation of the six sections of the Uinta Fm. (Townsend, Friscia & Rasmussen, 2006). (B) Minimum number of A. pulchra individuals. Green rectangle corresponds with meter level range for WU-34 (226–248 m). (C) Correlation of the measured stratigraphic section of Townsend, Friscia & Rasmussen (2006) relative to the Global Magnetic Polarity Time scale, using magnetostratigraphic section of Townsend et al. (2010) and Prothero (1996).

The Uinta Fm. is the highly fossiliferous type formation of the Uintan NALMA (Wood et al., 1941; Prothero, 1996) (Figs. 1 and 2A). The study area lies between latitudes 40°00′ and 40°30′ north and longitudes 109°00′ and 109°45′ west (Townsend, Friscia & Rasmussen, 2006) (Fig. 1). Most of the localities discussed here are tied to a stratigraphic section described by Townsend, Friscia & Rasmussen (2006) that extends 366 m through the older Uinta B (0–137 m) into the younger Uinta C (140–366 m), resulting in the first known conformable contact between the Uinta and Duchesne River Formations at 366 m (Osborn, 1895; Osborn, 1929; Prothero, 1996; Townsend, Friscia & Rasmussen, 2006) (Fig. 2A). Gunnell et al. (2009) divided the Uintan NALMA into four biochronological zones (Ui1a, Ui1b, Ui2, Ui3) on the basis of mammalian biostratigraphy of the Uinta, Bridger, and Washakie Formations. Material in the current study occurs in the immediate area of the stratotype localities for biochrons Ui2 and Ui3 or can be stratigraphically correlated with them (Gunnell et al., 2009; Townsend et al., 2010; Smith et al., 2017; Smith et al., 2020; Stidham, Townsend & Holroyd, 2020) (Fig. 2).

Only one turtle (Baena inflata) is reported from Uinta A, while Uinta B and C combined contain all other reported taxa (Gilmore, 1916). Baena inflata has been grouped with “Baena” affinis (Leidy, 1871), which was reestablished by Joyce & Lyson (2015), but a recent survey of Uintan baenids was unable to find additional material referable to the species (Smith et al., 2017). Uinta A has often been mistaken for the lower levels of Uinta B, and many workers have concluded that the lowest approximately 150 m of the formation does not bear fossils (Osborn, 1895; Riggs, 1912; Osborn, 1929; Prothero, 1996).

Materials & Methods

We used measured stratigraphic sections from Townsend, Friscia & Rasmussen (2006), which were recorded during the summers of 1997, 1998, 2000, and 2014. Fossil collection and stratigraphic work was conducted in a restricted area of the eastern Uinta Basin, on public land administered by the Bureau of Land Management (Paleontological Resources Use Permit Number UT06-031S). This study also includes published specimens from the Carnegie Museum of Natural History and the Yale Peabody Museum of Natural History and examines previously unpublished specimens from Brigham Young University Museum of Paleontology, the Natural History Museum of Utah, and the Utah Field House of Natural History State Park Museum. Collections from the latter three museums were integrated into the measured stratigraphy of Townsend, Friscia & Rasmussen (2006) from locality data on file at each repository. Additional records have been included from the University of California Museum of Paleontology from elsewhere in the basin, but these cannot be included in the detailed stratigraphic framework. Measurements of fossil specimens were taken using Mitutoyo Absolute Digimatic digital calipers, and from high quality digital images using ImageJ software (Rasband, 1997-2016). Magnified photos were produced using an Olympus SZX7 stereo microscope. Unless otherwise specified, all measurements are in millimeters (mm), recorded to the nearest 0.01 mm and rounded to the nearest 0.1 mm. Nomenclature for vertebral scales conforms to that proposed by Danilov et al. (2017).

Anatomical Abbreviations

The following anatomical abbreviations are used: co, costal; ne, neural; nu, nuchal; pe, peripheral; py, pygal; sp, suprapygal.

Systematic paleontology

TESTUDINES Batsch (1788)	
CRYPTODIRA Cope (1868)	
TRIONYCHIA Hummel (1929)	
CARETTOCHELYIDAE Gill (1889)	
ANOSTEIRALeidy (1871)	
Anosteira pulchraClark (1932)	
Figs. 3–6; Tables 1–2	
Synonymy. Pseudanosteira pulchraClark (1932)	

Holotype. CM 11808, a complete carapace, nearly complete hyoplastra, hypoplastra, and anterior extremities of posterior plastral lobe.

Figure 3 Carapace material of Anosteira pulchra from the Uinta Fm.

(A) Dorsal and (B) ventral views of UMNH.VP.27632, an articulated nuchal and left peripheral 1. (C) Right lateral view of UMNH.VP.31059, an articulated neural 3 and 4. (D) Right lateral, (E) dorsal, and (F) ventral views of UMNH.VP.27146, an articulated partial carapace. (G) Dorsal, (H) ventral, and (I) left lateral views of UMNH.VP.30590, a neural 6 and 7. (J) Dorsal, (K) ventral, and (L) left lateral views of UMNH.VP.30590, a suprapygal and pygal. (M) Dorsal, and (N) ventral views of UNMH.VP.19951, a right costal 1. (O) Dorsal, (P) ventral, and (Q) posterior views of UMNH.VP.31058, a right peripheral 2. (R) Dorsal, (S) ventral, and (T) anterior views of UMNH.VP.27077, a left peripheral 3. (U) Dorsal, (V) ventral, (W) medial, and (X) posterior views of UMNH.VP.27077, a left peripheral 6. (Y) Dorsal, (Z) ventral, and (AA) anterior views of UMNH.VP.30590, a right peripheral 8. Dotted black lines indicate edges of missing bone, vertical blue lines indicate orientation of the midline, and purple lines indicate sulci. All parts of figure to same scale.

Newly Referred Specimens. Table 1 contains 95 previously undescribed specimens recovered from the measured stratigraphic section of Townsend, Friscia & Rasmussen (2006). The minimum numbers of individuals, based on the maximum number of individual elements at each locality, is 37 (see Discussion).

Figure 4 Plastral material of Anosteira pulchra from the Uinta Fm.

(A) Ventral, and (B) dorsal views of UMNH.VP.19551, a partial left plastron. (C) Ventral, and (D) dorsal views of UMNH.VP.27452, a nearly complete left hypoplastron. (E) Ventral, (F) medial, and (G) dorsal views of UMNH.VP. 26554, a partial left hypoplastron. (H) Ventral, and (I) dorsal views of UMNH.VP.26917, a partial right hypoplastron with probable rodent gnaw marks circles in red. (J) Ventral, (K) dorsal, (L) medial, and (M) lateral views UMNH.VP.20525, a nearly complete right xiphiplastron. Dotted black lines indicate edges of missing bone and vertical blue lines indicate orientation of the midline. All parts of figure to same scale.

Type Locality and Horizon. Quarry L, Leota Ranch, near village of Ouray, Uinta County, Utah, USA (Clark, 1932, Fig. 7). Upper Horizon C (Clark, 193: 161), Uinta Formation, Lutetian, middle Eocene.

Description

Due to the large sample size in this study, the specimens described below were selected as representative elements of A. pulchra found within the measured stratigraphic section of Townsend, Friscia & Rasmussen (2006).

Figure 5 Magnified ventral surface of hypoplastral fragment UMNH.VP.26917, showing traces of rodent incisors (indicated by arrows) near the hypo-xiphiplastron suture.

Scale shows 1 mm increments and black arrows indicate orientation.

Figure 6 Associated carapace and plastron of Anosteira pulchra, specimen UMNH.VP.31072.

(A) Vertebral series and suprapygal in dorsal view. (B) Plastron and peripheral ring in dorsal view. (C) Vertebral series and suprapygal in ventral view. (D) Plastron and peripheral ring in ventral view. All parts of figure are at same scale. Vertical blue lines indicate orientation of the midline.

Carapace (Fig. 3)

UMNH.VP.27632 is an anterior carapace margin that includes the nuchal and left first peripheral (Figs. 3A–3B). There is a midline protuberance approximately 7 mm wide and 5 mm long that is raised 1.5 mm above the dorsal surface of the carapace, occupying most of the midline space between the anterior free margin and the intervertebral sulcus between the fused cervical/vertebral 1 and vertebral 2 scales (Fig. 3A). The protuberance forms the anterior limit of the dorsal keel, and a rounded dorsal projection is the most robust point along the thickened margin of the nuchal embayment (Fig. 3A). The anterior extremities of the sulci forming the slightly sigmoidal lateral sides of vertebral scale 2 project posteriorly from the aforementioned intervertebral sulcus (Fig. 3A). The sulci of this element are generally thin (<0.5 mm) and finely incised (Fig. 3A). Dorsal surface sculpture consists of a network of grooves that are roughly parallel to the free margin of the carapace (Fig. 3A). Grooves are shorter, more clustered, and have more pronounced relief where the periphery changes direction, as at peripheral 1 (Fig. 3A). The dorsal surface is quite smooth near the midline of the nuchal, where a slight ridge indicates the beginning of the median keel (Fig. 3A). The ventral surface of UMNH.VP.27632 is smooth except for finely toothed sutures between the specimen and adjacent bones (Figs. 3A–3B). A pair of gracile projections extend from the internal surface of the carapace to articulate with cervical vertebra 8 (Fig. 3B). Each projection is approximately 2.4 mm wide, 1 mm long, and 1.7 mm tall, crescent-shaped, and concave posteriorly (Fig. 3B).

Table 1 Uinta Fm. Anosteira specimens by stratigraphic meter level.

* indicates a BYU locality that is not assigned a meter level.

Specimen	MWU locality	Meter Level	Element	
UMNH.VP.27635	WU-123	366	Shell fragments	
UMNH.VP.27634	WU-49	364	Neurals; many shell fragments	
UMNH.VP.27212	WU-49	364	Shell fragments	
UMNH.VP.27077	WU-50	361	Left peripherals 3, 6	
UMNH.VP.27202	WU-50	361	Left peripheral 7; right hypoplastron fragment; articulated right nuchal/peripheral 1	
UMNH.VP.27146	WU-50	361	Partial left hypoplastron; right peripherals 1-2, possible 4, 10; neurals 2-4, 6; costals 3-5	
UFH 2002.19.2	WU-185	334	Partial carapace including neural	
UFH 2002.19.3	WU-185	334	Shell fragments	
UMNH.VP.27299	WU-223	332	Pygal	
UMNH.VP.27307	WU-223	332	Right peripheral 6, 8, 10; pygal; possible left hyoplastron fragment; partial right xiphiplastron; 1 possible right hypoplastral fragment	
UMNH.VP.26539	WU-223	332	Left peripherals 5-6	
UMNH.VP.26917	Above WU-216	286	Right hypoplastron fragment	
UMNH.VP.26919	Above WU-216	286	Suprapygal	
UMNH.VP.26504	Above WU-216	286	Partial pygal; partial peripheral	
UMNH.VP.26920	Above WU-216	286	Plastron fragment	
UMNH.VP.26511	Above WU-216	286	Carapace fragments	
UMNH.VP.18945	WU-45	285	Plastron and carapace fragments	
UMNH.VP.20505	WU-216	284	Right peripherals 1, 6-7; partial neural; costal fragments	
UMNH.VP.20506	WU-216	284	Partial hypoplastron	
UMNH.VP.20518	WU-216	284	Carapace fragments	
UMNH.VP.20498	WU-216	284	Pygal; costal fragments; posterior hypoplastron	
UMNH.VP.20479	WU-216	284	Carapace fragments	
UMNH.VP.20496	WU-216	284	Partial nuchal; partial costal; partial hyoplastron	
UMNH.VP.20525	WU-216	284	Partial costals; left peripherals 1-6, right peripherals 4-6; pygal; right xiphiplastron	
UMNH.VP.20523	WU-216	284	Right peripheral 6	
UMNH.VP.20522	WU-216	284	Right peripheral 6	
UMNH.VP.20532	WU-216	284	Carapace fragments	
UMNH.VP.20533	WU-216	284	Carapace and plastron fragments	
UMNH.VP.20535	WU-216	284	Carapace and plastron fragments	
UMNH.VP.20536	WU-216	284	Carapace and plastron fragments	
UMNH.VP.20537	WU-216	284	Carapace and plastron fragments	
UMNH.VP.20538	WU-216	284	Carapace and plastron fragments	
UMNH.VP.20539	WU-216	284	Carapace and plastron fragments	
UMNH.VP.20540	WU-216	284	Carapace and plastron fragments	
UMNH.VP.20541	WU-216	284	Carapace and plastron fragments	
UMNH.VP.20542	WU-216	284	Carapace and plastron fragments	
UMNH.VP.20543	WU-216	284	Carapace and plastron fragments	
UMNH.VP.20551	WU-216	284	Carapace and plastron fragments	
UMNH.VP.20552	WU-216	284	Carapace and plastron fragments	
UMNH.VP.20553	WU-216	284	Carapace and plastron fragments	
UMNH.VP.17724	WU-121	282	Carapace fragments	
UMNH.VP.30592	WU-134	226-248	Partial peripherals; small fragments	
UMNH.VP.30593	WU-134	226-248	Small fragments	
UMNH.VP.30594	WU-134	226-248	Partial peripherals; many small fragments	
UMNH.VP.30595	WU-134	226-248	Left peripherals 5, 6, 8; plastron fragment	
UMNH.VP.27424	WU-134	226-248	Pygal; partial peripherals; shell fragments	
UMNH.VP.20582	WU-134	226-248	Carapace fragments	
UMNH.VP.20583	WU-134	226-248	Carapace fragments	
UMNH.VP.20584	WU-134	226-248	Carapace fragments	
UMNH.VP.30596	WU-134	226-248	Costal fragments; peripherals	
UMNH.VP.30597	WU-134	226-248	Neurals 2-3; plastron fragments	
UMNH.VP.30598	WU-134	226-248	Pygal; peripheral fragments; carapace fragments; plastron fragments	
UMNH.VP.30599	WU-134	226-248	Neural; peripheral fragments	
UMNH.VP.30600	WU-134	226-248	Neural 5 or 6; right peripherals 5-6; left peripherals 3-6; left possible hyoplastron fragment; anterior peripherals; carapace fragments; plastron fragments	
UMNH.VP.30602	WU-134	226-248	Left peripheral 5; left possible hypoplastron fragment; indet. plastron fragment.	
UMNH.VP.30603	WU-134	226-248	Costals; neurals	
UMNH.VP.30604	WU-134	226-248	Articulated partial anterior carapace including nuchal	
UMNH.VP.30605	WU-134	226-248	Neurals 2-4; anterior peripheral; partial peripheral; many tiny fragments	
UMNH.VP.27450	WU-134	226-248	Peripheral; shell fragments	
UMNH.VP.27452	WU-134	226-248	Pygal; left hypoplastron	
UMNH.VP.30586	WU-134	226-248	Many small fragments	
UMNH.VP.30587	WU-134	226-248	Many costal fragments	
UMNH.VP.30588	WU-134	226-248	Partial left hypoplastron	
UMNH.VP.30589	WU-134	226-248	Partial peripherals; small fragments	
UMNH.VP.30590	WU-134	226-248	Right peripheral 8, neurals 6-7, pygal, suprapygal	
UMNH.VP.30591	WU-134	226-248	Left and right peripheral 1	
UMNH.VP.30910	WU-134	226-248	Neurals 2-3	
UMNH.VP.27226	WU-134	226-248	Small fragments (mostly plastron)	
UMNH.VP.27453	WU-134	226-248	Partial pygal; partial nuchal; partial peripherals; small fragments	
UMNH.VP.27630	WU-134	226-248	Plastral fragments	
UMNH.VP.27454	WU-134	226-248	Right xiphiplastron fragment	
UMNH.VP.27632	WU-134	226-248	Nuchal; left peripheral 1	
UMNH.VP.26515	WU-26	237	Many small plastron fragments	
UMNH.VP.26554	WU-26	237	Neurals 1-3; partial left hypoplastron; probable femora; partial peripherals; many tiny fragments	
UMNH.VP.31070	WU-26	237	Partial peripherals; many fragments	
UMNH.VP.31058	WU-26	237	Peripheral 2; partial costals; small fragments	
UMNH.VP.31059	WU-26	237	Neurals 3-4; small fragments	
UMNH.VP.31060	WU-26	237	Partial peripherals; small fragments	
UMNH.VP.26556	WU-26	237	Bridge peripherals	
UMNH.VP.19951	WU-12	141	Right costal 1	
UMNH.VP.27281	WU-1	106	3 possible individuals; partial peripherals; shell fragments; 3 pygals; right peripheral 1	
UMNH.VP.20034		*	Shell fragments	
UMNH.VP.20405		*	Partial hypoplastron, partial costal	
UMNH.VP.20231		*	Plastron and carapace fragments	
UMNH.VP.30607	WU-54	96	Peripheral 2	
UMNH.VP.30606	WU-54	96	Posterior peripherals	
UMNH.VP.30601	WU-54	96	Bilateral hyoplastra, indeterminate partial costal, 40 carapace fragments	
UMNH.VP.18943	WU-32	>95	Plastron and carapace fragments	
UMNH.VP.18935	WU-32	>95	Plastron and carapace fragments	
UMNH.VP.20661	WU-32	>95	Right peripherals 6, 7	
UMNH.VP.27306	WU-23	∼83	Left and right peripheral 5; posterior peripheral fragments	
UMNH.VP.31072	WU-8	57-60	Associated partial carapace and plastron	
UMNH.VP.31073	WU-8	57-60	Pygal	
UMNH.VP.27243	WU-18	25	2 individuals; partial peripherals; plastron fragments; pygals; left hypoplastron; indeterminate shell fragments	

Table 2 Anosteira pulchra records from the Uinta Fm., outside of the measured stratigraphic section of Townsend, Friscia & Rasmussen (2006).

Specimen	Locality	Element	
UCMP 218731	V98069	Shell fragments	
UCMP 223356	V98069	Hyo- or hypoplastral fragment	
UCMP 223357	V98069	Hyo- or hypoplastral fragment	
UCMP 223358	V98069	Bridge peripheral	
UCMP 223359	V98069	Peripheral	
UCMP 223360	V98069	Peripheral	
UCMP 223361	V98069	Peripheral	
UCMP 235587	V98069	Bridge peripheral	
UCMP 235588	V87136	Left hyoplastron and shell fragments	
UCMP 223098	V71057	Peripheral 2	
UCMP 223099	V71057	Peripheral 8	
UCMP 218732	V71058	Shell fragments	
UCMP 223355	V71058	Shell fragments	

UMNH.VP.31059 (Fig. 3C) and UMNH.VP.27146 (Figs. 3D–3F) are partial anterior neural rows of A. pulchra, with a characteristic anterior spike in the midline carina (keel) arising from neurals 3 and 4 (Figs. 3C–3D). The spike falls sharply in the posterior third of neural 4, returning to approximately the same maximum height as the midpoint of neural 4 (Figs. 3C–3D). In dorsal and ventral views of UMNH.VP.27146, neural 2 is pentagonal and uniformly wide, and neurals 3-6 are hexagonal, wider anteriorly, and have short anterior sides (Figs. 3E–3F). Neural 5 of UMNH.VP.27146 is missing (Figs. 3D–3F), though the keel of neural 6 was likely similar in height (Fig. 3D).

Figure 7 Scale pattern variation within Anosteira pulchra.

(A) Dorsal carapace of CM 11808, type specimen of A. pulchra. (B) Detail of carapacial scale pattern of CM 11808 as previously published (Clark, 1932), with yellow star indicating unmarked region of shell. (C) Dorsal carapace of YPM VPPU 16318, mentioned in Joyce (2014). (D) Detail of carapacial scale pattern of YPM VPPU 16318. (E) Dorsal carapace of YPM VPPU 16317, mentioned in Havlik, Joyce & Böhme (2014) and Joyce (2014). (F) Detail of carapacial scale pattern of YPM VPPU 16317. (G) Partial carapace with scale pattern of UMNH.VP.27146. (H) Scale pattern of neural spike of larger individual in dorsolateral view of UMNH.VP.27453. (I) Scale pattern of third neural of smaller individual in dorsolateral view of UMNH.VP.27453. (J) Scale pattern of partial carapace of UMNH.VP.31072 in dorsal view. Red lines indicate sulci and black lines indicate sutures. Photos of YPM specimens courtesy of Yale Peabody Museum of Natural History (https://collections.peabody.yale.edu/search/).

UMNH.VP.30590 (Figs. 3G–3L) consists of associated posterior midline elements (neurals 6 and 7, suprapygal, and pygal), as well as peripheral 8 described below (Figs. 3Y–3AA). Neural 6 is generally rectangular dorsally, measuring 7.5 mm long and 4.2 mm wide (Figs. 3G–3H). Neural 7 is proportionally shorter, and is 8.1 mm long and 6.2 mm wide (Figs. 3G–3H). The dorsal outline of neural 7 is distinctly hexagonal, and its surface area is larger dorsally than ventrally (Figs. 3G–3H). Both posterior neurals have a smooth dorsal surface, and the posterior keel of neural 6 is warped slightly laterally (Fig. 3G). The keel of neural 6 is triangular in profile and forms a second spike behind that of neural 4, rising approximately 3 mm above the external surface (Fig. 3I). Midline parts of UMNH.VP.30590 are missing between the posteriormost neurals and suprapygal (Figs. 3G–3L). The eighth costals are missing, but meet at the midline in situ in complete specimens (see Clark, 1932). A tightly beaded pattern covers the dorsal and ventral surfaces of the pygal posterior to the anterior ventral embankment (Figs. 3J–3L). The posterior pygal margin is acute, similar to the posterior peripherals, but is thickest at the midline (Figs. 3J–3L and 3AA). The pygal has a midline sulcus along the dorsal surface, as described above (Fig. 3J). A low keel bisects the suprapygal along the dorsal midline, and the ventral surface of the suprapygal is smooth and slightly concave (Figs. 3J–3K). The suture between the suprapygal and pygal is finely dentate (Fig. 3K), and the pygal flares posteriorly and dorsally (Figs. 3J–3L).

UMNH.VP.19951 is a right costal 1 that is missing two sections of its posterior edge (Figs. 3M–3N). It has a length of 21.9 mm and a width of 41.2 mm. Its posterior suture is concave anteriorly, and its anterior margin convex, where it is sutured for articulation with the nuchal and the first three peripherals (Figs. 3M–3N). The medial and lateral sutures are preserved, indicating articulation with neural 1 and the anterior portion of peripheral 3, respectively (Fig. 3N). The bone is thinnest near its middle, and the head of the first rib is separated from the medial suture and flanked by several small foramina (Fig. 3N). Otherwise, the ventral surface is smooth, and the dorsal surface shows little evidence of texture apart from a few oblong pits and small gouges (Fig. 3M).

UMNH.VP.31058 is a right peripheral 2 that has the characteristic flattened cylindrical shape of the anteriormost peripherals (Figs. 3O–3Q). Its lateral edge is straight (Figs. 3O–3P), and the lateral margin is rounded in cross section (Fig. 3Q). No sulci are present, and a finely pitted texture is present only in dorsal view (Fig. 3O). The surface becomes smooth along the lateral edge and ventral view of the bone (Fig. 3P).

UMNH.VP.27077 is a left peripheral 3 that is missing its anteromedial corner (Figs. 3R–3S). Its ventral surface is smooth (Fig. 3S), and its dorsal surface is slightly rugose and damaged by two large, irregular pits near the lateral edge (Fig. 3R). The posterolateral margin projects ventrally and there are two prominent sockets that mark articulation with the hyoplastron and the beginning of the bridge series of peripherals (Fig. 3S). The anterior half of the lateral margin maintains the flattened cylindrical character of the peripherals anterior to it, but the edge slopes sharply ventrally as it forms the seat of the axillary buttress of the bridge (Fig. S-T).

UMNH.VP.27077 also includes a left peripheral 6 with robust gomphotic sockets that characterize bridge peripherals (Figs. 3U–3X). Anteriorly, peripherals are thin and rod-like (Figs. 3O–3Q), become thick and triangular in the bridge region (Figs. 3R–3X), and are wide and flat posteriorly (Figs. 3Y–3AA). Peripheral 8, associated with other elements from UMNH.VP.30590 described above (Figs. 3G–3L), is an example of the broad, flat, acutely-margined posterior peripherals (Figs. 3Y–3AA). It is 19.2 mm long, 18.1 mm wide, and 9.77 mm tall, and only its dorsal surface is sculptured (Fig. 3Y). An intermarginal sulcus crosses the dorsal surface transversely at its anterior third (Fig. 3Y), and a longitudinal, rounded embankment tapers posteriorly along the medial side of the ventral surface (Fig. 3Z).

To summarize, peripherals articulate to form a slightly flaring, often scalloped ring whose most distal parts are thin and delicate (Figs. 3Y–3AA, 6B and 3D). Distinct gomphoses indicate clear articulations between bridge peripherals 3–7 and adjacent bones of the carapace and plastron (Figs. 3T and 3X), while anterior peripherals 1–2 and posterior peripherals 8–10 only articulate with the carapace (Figs. 3Q and 3AA). The angle formed by the dorsal and ventral faces at the lateralmost edge of the shell is approximately 66.5° in peripheral 6 (Fig. 3X), but becomes acute to approximately 28° in the posterior peripherals (Fig. 3AA). A distinct median dorsal carina (keel) forms a blunt, posteriorly-oriented spike on neurals 3–4 (Clark, 1932) (Figs. 3C–3E). The carina continues posteriorly and terminates on the antero-dorsal view of the pygal as a distinctly raised midline ridge anterior to the confluence of the marginal scales (Fig. 3J). The pygal is robust and trapezoidal (Figs. 3J–3K). It has a pronounced embankment perpendicular to the midline in antero-ventral view, as in all carettochelyids, forming a posterior wall of the body cavity (Havlik, Joyce & Böhme, 2014; Joyce, 2014) (Figs. 3K–3L).

Plastron (Fig. 4)

UMNH.VP.19551 is an articulated left hyo- and hypoplastron that helps form a classic reduced “cruciform” plastron (Figs. 4A–4B). It is missing a portion of the anteromedial corner of the hypoplastron, and the anterior and posterior parts of the bridge region (Figs. 4A–4B). The maximum length of the specimen is 31.9 mm, of which 18.5 mm accounts for the hypoplastron. Its overall maximum width is 40.7 mm, and the hypo-xiphiplastral suture is 9.2 mm wide. The bridge region is flattened and the hypoplastron is longer than the hyoplastron at their narrowest points (Figs. 4A–4B). The ventral surface is smooth near the midline and rugose at the middle of the specimen, with parallel striations projecting toward the bridge articulation (Fig. 4A). The dorsal surface is smooth except for short grooves near the bridge and raised red concretions in the hyo-hypoplastral suture (Fig. 4B). The anterior edge of the hyoplastron forms a rounded “M” shape, with larger medial and smaller lateral, anteriorly-projecting projections that form the seat for the epiplastron (Figs. 4A–4B). The medial projection is finely pitted along its anterior edge, likely for ligamentous attachment to the epiplastron and entoplastron (Figs. 4A–4B). It is notable that the hypo-xiphiplastral suture of UMNH.VP.19551 (Figs. 4A–4B) is relatively straight, compared with the sinusoidal sutures of the specimens described below, though this may be attributable to breakage (Figs. 4C–4I).

UMNH.VP.27452 is a nearly complete left hypoplastron (Figs. 4C–4D). The bridge region is fractured at its narrowest, central point (8.8 mm wide) (Figs. 4C–4D). The hyo-hypoplastral suture is visible along the bone’s anteromedial edge, where the bone is thinnest (2.9 mm) (Figs. 4C–4D). The sutures of this area are better preserved in the smaller left hypoplastron UMNH.VP.26554 (Figs. 4E–4G) and the sutures shared with adjacent bones are intact (Figs. 4E–4G). In UMNH.VP.26554, the hyo-hypoplastral suture and the midline form an approximately 73° angle (Fig. 4E and 4G). The width of the left hypo-xiphiplastral suture is 12.39 mm and the plastron has a maximum thickness of 6.2 mm (Figs. 4E–4G). The partial right hypoplastron UMNH.VP.26917 is 24.2 mm long and 14.8 mm wide (Figs. 4H–4I). Its ventral surface has perhaps the clearest defined texture of all the plastra examined in this study (Fig. 4H). On it, there is a series of four distinct, nearly parallel trace marks on the ventral surface of UNMH.VP.26917, immediately anterior to the hypo-xiphiplastral suture (Figs. 4H and 5). These are are shown magnified in Fig. 5, interpreted and discussed below.

UMNH.VP.20525 is a nearly complete right xiphiplastron that is 32.2 mm long and 11.5 mm wide (Figs. 4J–4M). The bone is narrow and its lateral edge is nearly parallel to the midline, but its posterior quarter tapers to a point (Figs. 4J–4K) indicating the lack of anal notch as in other Anosteira spp. The hypo-xiphiplastral suture is sinusoidal, and the articular surface along the suture is comprised of a complex network of gomphotic scarph pegs and sockets (Figs. 4J–4M). It is generally even in thickness, but is thickest anteriorly along the midline (Fig. 4L). The bone bends dorsally and its posterior point forms a distinct spike with several longitudinal ridges on the dorsal surface (Fig. 4K). Both the dorsal and ventral surfaces are mostly smooth, and several small foramina are present in the anterior half of the dorsal side (Fig. 4K). A narrow groove runs along the posterior end of the lateral side of the bone, which is thinnest near its middle (Fig. 4M). This groove probably marks the limit of the skin contact on the dorsal surface.

An associated carapace and plastron (UMNH.VP.31072) (Fig. 6)

One specimen from the current sample has been recovered with an associated carapace and plastron (Fig. 6). The carapace consists of a mostly complete neural row, including neurals 2–6 and adjacent costals (Figs. 6A and 6C ), along with a peripheral ring that is missing only the left peripheral 3, right peripheral 5, and significant portions of bilateral peripherals 4 and 8 (Figs. 6B and 6D). Neurals 1 and 7 are missing, though most of the suprapygal is preserved including its midline keel (Figs. 6A and 6C). Apart from the medial portions which articulate with the neural series (Figs. 6A and 6C), the costals were fractured into dozens of tiny fragments from the middle of the bones.

The plastron of UMNH.VP.31072 is well preserved, missing only the anterior half of the right xiphiplastron, approximately the posterior third of the left xiphiplastron, and lateral portions of the bilateral hyoplastra (Figs. 6B and 6D). The anterior plastral lobe is represented by one fragment of the epiplastron which articulates with the curved anteromedial margin of the hyoplastron (Figs. 6B and 6D). This posterior portion of the right epiplastron is thickest along a ridge at the middle of the width of the bone, and a narrow groove lies along the medial side of the ridge (Fig. 6B). There are fine striations near the midline, anterior to the groove, possibly indicating ligamentous articulation associated with the kinetic hinge at the epi-hyoplastral contact (Fig. 6B). The remainder of the plastron is consistent with the specimens described above, and the preserved right xiphiplastron tapers to a thickened point posteriorly, as in UMNH.VP.20525 (Figs. 4J–4M). This specimen is the most complete individual of Anosteira pulchra in the current study and allows a simple estimation of the turtle’s size. Using relative proportions from the type specimen (CM 11808) (Fig. 7A), UMNH.VP.31072 is estimated to have a midline carapace length of 15.3 cm, approximately 80% the size of CM 11808.

Results

The 95 Anosteira pulchra specimens in this study (Table 1) substantially increase the sample of this taxon and provide new insights into its stratigraphic distribution in the Uinta Formation, which are discussed below. Uinta C contains most occurrences and the stratigraphic range of the species is extended into older Uinta B sediments (Fig. 2). Additional Uintan records of Anosteira pulchra from outside the study area are provided in Table 2. This set of specimens cannot be correlated with the measured stratigraphy of Townsend, Friscia & Rasmussen (2006), but they demonstrate the presence of A. pulchra in other parts of the Uinta Basin, suggesting areas worthy of further collecting and stratigraphic analysis. UCMP locality V98069 is near Starvation Reservoir (Duchesne County, UT) and is partially surrounded by Uinta B and C strata (Sprinkel, 2018) (Fig. 1). Localities V71057 and V71058 are northwest of Ouray (Uintah County, UT), near Myton Pocket, and V98069 is near the study area, but not MWU localities (Sprinkel, 2007) (Fig. 1).

Discussion

Distribution of Anosteira pulchra in the Uinta Formation

Two major facies can be described for both the lower and upper intervals of the Uinta Fm. stratigraphic section. The lower intervals are typified by mud and claystone over-bank deposits near fine-grained channel sandstones, with very little soil development (0–∼140 m; Townsend, Friscia & Rasmussen, 2006). The upper intervals (140–366 m) are characterized by more mature paleosols, interspersed with channel sandstones composed of larger clasts and stones, as well as some ponds (WU-26; Westgate et al., 2013). Specimens of Anosteira pulchra are found in both facies types, and numerous specimens were recovered from both WU-134 and WU-26, localities in the upper intervals of the formation (Fig. 2B). Westgate et al. (2013) determined that the WU-26 locality was very likely a pond, and although extensive sedimentological study has not been performed at WU-134, a series of mature paleosols are present and further work may reveal that this was also a pond site. More specimens are certainly associated with the upper intervals of the formation and localities with mature paleosol development and there are fewer specimens from the lower intervals. It is not possible to determine if this difference is an ecological preference of A. pulchra, or if it is a taphonomic bias due to less over-bank flooding during the time of paleosol development preserved in the upper intervals of the formation, which allowed for greater accumulation of skeletal elements.

Historically, most collecting in the Uinta Fm. has focused on mammals, and the most frequently collected and most productive fossil mammal localities occur near the top and bottom of the section (Townsend, Friscia & Rasmussen, 2006; Townsend et al., 2010) (Fig. 2A). It is noteworthy that nearly all of the specimens collected and examined in this study were collected from the surface or by traditional excavation techniques. Material from four locations at approximately 280 m (Fig. 2A) was screenwashed but produced no turtle fossils. Since 2007, more than 25 tons of bulk sample have been excavated from deposits at 237 m (Murphey et al., 2017). This work has yielded more than 400 mammal specimens identifiable to genus or species (Westgate et al., 2013). Only one Anosteira pulchra specimen (UMNH.VP.26554) was recovered via these means, providing additional evidence that the taxon is uncommon or patchy in distribution, rather than common and under sampled.

The minimum number of the individuals (MNI) calculated from the 95 Anosteira pulchra specimens reported in this study is 37, based on the maximum number of individual elements at each locality. Of the MNI, 78% occur above 140 m, in Uinta C sediments (Figs. 2A–2B). The maximum abundance occurs near 237 m, stratigraphically between the Glen Bench Bed and Sherbet Orange Bed (Figs. 2A–2B). The most significant gap is between the base of this interval (226 m) and the Uinta B-C boundary (137–140 m) (Fig. 2A). This interval contains the upper H section strata (below 200 m), which includes the Ruby Red Wash, Red Wash Yellow, and Susan’s Stripe Gray Marker Beds (Fig. 2A). The remaining 22% of the MNI were found in Uinta B rocks, without a substantial peak as in higher strata. Occurrences of A. pulchra in Uinta B are more evenly distributed and have lower abundances than Uinta C. A gap in the uppermost Uinta B sediments near Devil’s Playground 1 (106–137 m) is notable because this interval includes WU-117, a highly productive and well-sampled locality in the area. This suggests that the absence of A. pulchra fossils in the interval is not simply collection bias. Currently we have no sedimentologic explanation for the lack of A. pulchra in this 31 meter interval. Additional targeted collection in the future may reduce gaps, identify factors related to abundance, and clarify the trends reported here.

Evidence of rodent gnaw marks on UMNH.VP.26917

A hypoplastral fragment (UMNH.VP.26917) from 286 m (Uinta C) has four sets of linear excavations in the posterior half of its ventral surface, near the hypo-xiphiplastral suture (Figs. 4H and 5). The shell fragment is 24.4 mm long and 14.8 mm wide, consistent with the size of an adult turtle (Fig. 4H). Each of the scratches has a thin puncture at its lateral end and several associated scrape marks which travel anteromedially across the bone to a maximum of 7.8 mm (Fig. 5). The scrape components are approximately perpendicular to the punctures and the ornamental ridges of the bone, nearly parallel and without intersection (Fig. 5). Scrapes are deepest near to the puncture and gradually become shallow medially, indicating they were initiated laterally. The middle two punctures are most prominent, with shapes that are slightly sinusoidal and mirrored across the gap between them. The portions of the puncture nearest the gap are widest and deepest, penetrating the cortex. The anterior edges of each scrape are sharp and their floors rough, suggesting they had not undergone repair (Fig. 5).

The scratches are interpreted as gnaw marks inflicted by a rodent, consistent with compression punctures and tapering scratches described on Eocene turtles by Hutchison & Frye (2001). Rodent gnaw marks can be differentiated from those of carnivorans by their characteristic parallel series of furrows (Haglund, Reay & Swindler, 1988; Pobiner, 2008). The shape of the punctures indicates sharp flat teeth, consistent with rodent incisors, in addition to their small size (1.4–1.7 mm wide). The notable gap between the middle two foci (0.7 mm) suggests lower incisors, which are sometimes not immediately adjacent due to the unfused mandibular symphyses of rodents (Addison & Appleton, 1915; Weijs, 1975). Rodents were common in a variety of sizes in Uinta C of the Uinta Fm. (see Rasmussen et al., 1999), and the tracemaker was relatively small.

General remarks on shell structure and kinesis in Anosteira pulchra

The two North American species of Anosteira (A. ornata; Leidy, 1871 and A. pulchra; Clark, 1932) are distinguished from one another primarily by the shape of neurals and arrangement of vertebral scales (Hay, 1906; Clark, 1932; Hutchison, 1996; Joyce, 2014). Both species of Anosteira (Clark, 1932) have a broadly ovate carapace with a shallow nuchal embayment (Hay, 1908; Clark, 1932) (Figs. 3A–3B). The plastral morphology of Anosteira is similar to other trionychians, intermediate in size between the narrow, cruciform plastron of Kizylkumemys and the large plastron of the Carettochelyinae (Havlik, Joyce & Böhme, 2014; Joyce, 2014). The plastra of Anosteira spp. (and all Carettochelyidae) exhibit no visible sulci, indicating that no plastral scales were present (Havlik, Joyce & Böhme, 2014; Joyce, 2014) (Figs. 4, 6B and 6D). Unlike Trionychidae, Anosteira features scales and sulci on the carapace, and has ten pairs of peripherals (Havlik, Joyce & Böhme, 2014; Joyce, 2014) (Figs. 3 and 6). The periphery of A. pulchra forms a robust structural ring around the margin of the carapace (Figs. 6B and 6D). Sutures between adjacent peripherals are generally articulated via fine dentate sutures, but many sutures in the plastron show broader and more diffuse areas of soft tissue connection, indicative of kinesis. Kinesis was possible along the anterior edge of the hyoplastra, along the plastral midline suture, and at the hypo-xiphiplastral suture, a general pattern seen in other carettochelyids (e.g., Meylan, 1988; Meylan & Gaffney, 1989; Hutchison, Holroyd & Ciochon, 2004; Joyce et al., 2012) and more generally in highly aquatic turtles (Bramble, 1974; Bramble, Hutchison & Legler, 1984; Angielczyk, Feldman & Miller, 2010). The number of kinetic sutures and range of motion primarily enabled the head and neck to be withdrawn under the carapace. Some flattening of the shell and the accommodation of relatively enlarged fore flippers lateral to the shell were likely also permitted.

Vertebral scale pattern variation in Anosteira pulchra

In general, carettochelyids exhibit a wide variety of scale patterns between genera, species and even individuals, and the clade is sexual dimorphic in body size and posterior plastral kinesis (Joyce, Parham & Gauthier, 2004; Joyce et al., 2012; Joyce, 2014; Danilov et al., 2017). The partial carapace of UMNH.VP.27146 (Figs. 3E and 3E) provides a clear example of the most common scale pattern recovered in the current study. All published accounts of Anosteira pulchra (i.e., Clark, 1932; Gaffney, 1979; Havlik, Joyce & Böhme, 2014; Joyce, 2014; Danilov et al., 2017) are based on the holotype (CM 11808), which is a nearly complete carapace and plastron that is missing its entire anterior plastral lobe and most of the posterior lobe behind the hypo-xiphiplastral suture (Fig. 7A). CM 11808 has a pair of vertebral scales (the second and a coalesced third and fourth) that partly surround the anterior “additional vertebral” sensu Danilov et al. (2017). They are figured with a gap between them that occupies much of the length of costal 3 (see Clark, 1932) (Fig. 7B). An examination of the type specimen (CM 11808) reveals that Clark (1932) accurately figured the pattern traced on the type specimen in red (Fig. 7A). However, except for UMNH.VP.31072, all fossil material discussed in the current study repeats a pattern in which there is contact between vertebral scale 2 and combined vertebral scales 3 and 4 (Figs. 3E, 3C–3E). The degree of adjacency is apparently somewhat variable, as evident when comparing the pattern of UMNH.VP.27146 (Fig. 7E) with two well-preserved carapaces (YPM VPPU 016317 and 016318) from the 1936 Princeton Uinta Basin expedition (noted in Joyce, 2014) (Figs. 7C–7D). The scute pattern of UMNH.VP.31072 is notable for lacking contact between vertebral 2 and vertebrals 3 + 4 (as in the type), and asymmetrical constriction of the posterior extensions of vertebral 2 (Fig. 7H). However, contact between vertebrals 2 and 3 + 4 and sometimes slight lateral adjacency is the most frequently recovered variation (Figs. 7C–7E). While this study presents a modified scale arrangement from the type, it is consistent with the homology and resulting discussion of carettochelyid phylogeny in Danilov et al. (2017). It is unclear if the observed scale variation affected shell stability or is related to the broader carettochelyid trend of scale reduction and eventual loss. In any case, the longitudinal expansion of vertebral scales adjacent to the midline in A. pulchra is similar to that of A. ornata (Danilov et al., 2017). The variability in scalation we find in the Uintan A. pulchra and similarity of neural formula to A. ornata may suggest that this is a single species (as alluded to by Joyce (2014), Havlik, Joyce & Böhme (2014), and Joyce, Volpato & Rollot (2018) or a chronospecies. However, until larger samples of A. ornata from the Bridger Fm. can be described and any variation in these (or novel) features examined comparatively, we conservatively retain the two species. In total, this study provides a robust account of the morphology of A. pulchra, examines intraspecific variation of its vertebral scales, and expands its stratigraphic range into older Uintan strata. Future studies of stratigraphic distribution among the diverse turtle faunas of the Uinta Fm. may be useful in better understanding local and regional biostratigraphy during the Eocene.

Conclusions

The stratigraphic range of Anosteira pulchra in the Uinta Formation is demonstrated to extend into older Uinta B strata, rather than solely Uinta C. The vast majority of occurrences (78%) are concentrated in Uinta C localities with mature paleosol development, some with evidence of ponds, where they reach higher abundance than Uinta B localities. Current sedimentological data are insufficient to determine whether this is an ecological preference of A. pulchra, a result of changing climate during the late Uinta NALMA, a taphonomic bias associated with less over-bank flooding and greater skeletal accumulation in the uppermost intervals, or some combination of these factors. Considering A. pulchra material was present nearly to the bottom of the measured stratigraphic section, future collecting in even older intervals may yield additional insights. Further, given the similarities between the turtle assemblages of the Bridger and Uinta Formations, and the presence of A. pulchra in the intermediate Washakie Fm. (Joyce, Volpato & Rollot, 2018), additional taxonomic clarification may become possible through more collecting and study of existing material in institutional repositories.

A significant amount of intraspecific variation is apparent in A. pulchra, particularly in the shape of neurals and arrangement of vertebral scales. These are the primary diagnostic characters that distinguish A. pulchra from the older A. ornata, supporting previous hypotheses toward a single taxon. We suspect that this may indeed be true, however synonymy cannot be confirmed until the description of a larger Bridger Anosteira sample, and an assessment of the stratigraphic range of A. ornata are produced.

The authors wouldlike to thank Dr. Rodney Scheetz of Brigham Young University Museum of Paleontology, Dr. Carrie Levitt-Bussian of the Natural History Museum of Utah, Dr. Steve Sroka of the Utah Field House of Natural History State Park Museum, Amy Henrici (Section of Vertebrate Paleontology) of the Carnegie Museum of Natural History, and the Department of Vertebrate Paleontology at the Yale Peabody Museum of Natural History, for their assistance in accessing and photographing specimens. Additional thanks to Matt Kruback for photographic expertise on BYU specimens. Finally, we thank Drs. Walter Joyce, Ren Hirayama, and Adán Pérez-García for comments and suggestions that greatly improved the quality of the manuscript and figures.

Institutional abbreviations

BYU Brigham Young University Museum of Paleontology, Provo, Utah, USA

CM Carnegie Museum of Natural History, Pittsburgh, Pennsylvania, USA

MWU Midwestern University, Glendale, Arizona, USA

UCMP University of California Museum of Paleontology, Berkeley, California, USA

UFH Utah Field House of Natural History State Park Museum, Vernal, Utah, USA

UMNH.VP Vertebrate Paleontology Collection, Natural History Museum of Utah, Salt Lake City, Utah, USA

WU Washington University, St. Louis, Missouri, USA

YPM VPPU Princeton University collection in the Division of Paleontology, Yale Peabody Museum of Natural History, New Haven, Connecticut, USA

Additional Information and Declarations

Competing Interests

Author Contributions

Field Study Permissions

Data Availability

The authors declare there are no competing interests.

Brent Adrian conceived and designed the experiments, performed the experiments, analyzed the data, prepared figures and/or tables, authored or reviewed drafts of the paper, and approved the final draft.

Patricia A. Holroyd analyzed the data, prepared figures and/or tables, authored or reviewed drafts of the paper, and approved the final draft.

J. Howard Hutchison analyzed the data, authored or reviewed drafts of the paper, and approved the final draft.

KE Beth Townsend conceived and designed the experiments, analyzed the data, prepared figures and/or tables, authored or reviewed drafts of the paper, and approved the final draft.

The following information was supplied relating to field study approvals (i.e., approving body and any reference numbers):

Bureau of Land Management Paleontological Resources granted Use Permit Number UT06-031S, to Dr. Kathryn E. Townsend, Ph.D. Permission to study museum specimens was provided by: Brigham Young University Museum of Paleontology, Carnegie Museum of Natural History, University of California Museum of Paleontology, Utah Field House of Natural History State Park Museum, Natural History Museum of Utah, Salt Lake City, and Yale Peabody Museum of Natural History.

The following information was supplied regarding data availability:

Specimen numbers, locality and stratigraphic data, and descriptions for all specimens included in the study are available in Tables 1 and 2.

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
