# Peer review of "Additional records and stratigraphic distribution of the middle Eocene carettochelyid turtle Anosteira pulchra from the Uinta Formation of Utah, North America"

_PeerJ, doi:10.7717/peerj.9775_

## Round 0.1 · original submission · Minor Revisions

The reviewers are positive about your work, but still offer some suggestions for improvements. I note that currently most of the photos are suboptimal.

Please, together with your unmarked revised manuscript, provide a marked-up copy as well as a document explaining how you have addressed each of the points raised by the reviewers.

·

Basic reporting

see below

Experimental design

see below

Validity of the findings

see below

Additional comments

Dear Editors, Dear Authors,

this manuscript pertains to the description of new shell material of the somewhat enigmatic fossil turtle Anosteira pulchra from the Uintan of North America. The authors correctly highlight that this turtle has only been known by few previous occurrences. I am therefore very pleased to see that the authors were able to pull together so many specimens from different institutions to paint a more meaningful picture of this turtle. It is also wonderful to see that so many museums take their mandate so seriously of providing the scientific community access to unpublished public specimens in their care. I only wished this practice was more widespread.

All in all, this is a well-written manuscript that provides important new information. The geological section and stratigraphic work, of course, are outstanding. I nevertheless note some minor weaknesses that should perhaps be addressed:

1) I notice a few problems with the figures:

a) The labels and objects in Figure 3 are too small, I fear. In addition, keep in mind that PeerJ maintains that unnecessary white margin on the left side of each page, which means that the figure will become even more difficult to read once published, unless the authors request specifically that the figure is published at full page length, which is often forgotten during the proofing stage. I, therefore, suggest rearranging the figure into portrait format and using larger labels. Figure 7 could profit from this as well.

b) I know that the figures are a little fuzzy due to compression associated with the submission process, but it seems to me that some pictures have poor focal depth (e.g., Fig. 3D, F) or are simply out of focus (Fig. 3M, N, R, S, T). I know that it is difficult to photograph small material, but amazing results can be achieved with a relatively basic SLR camera in combination with a photo stand, basic lighting, and focus stacking (an option included in Photoshop, so no special software is needed). If the authors are willing to give it another try, but need advice, I will gladly send instructions. Incidentally, I cannot judge the quality of the images in Figure 4, as even the high-quality version is terribly pixelated. I understand that it may not be practical to photograph material held in distant collections, so this is not the highest priority, as the figures certainly reveal what they are supposed to show.

2) I am a little disappointed by the taxonomic aspects of this manuscript for several reasons:

a) The authors state that the two named species from North America have traditionally been distinguished by differences to their neural formula, vertebrals, and the dorsal spines, but I am unaware of any literature pertaining to the dorsal spines, as these have not yet been described for A. ornata (and were barely highlighted for A. pulchra by Clark, 1932). Please clarify or modify.

b) The authors failed to highlight that the new material negates purported differences in the neural formula between A. ornata and A. pulchra, as both species now seem to have hexagonal neurals with short anterior sides. The authors nevertheless note that differences exist in regards to "additional vertebral scales that surround the costo-neural region," but I have no idea what they are talking about. Indeed, as far as I can tell, there are no reasons left to conclude that two species are present. I already alluded to this in Joyce (2014), Havlik et al. (2014), and Joyce et al. (2018). I, therefore, would like to encourage the authors to either highlight more clearly what they believe to be the remaining differences, or synonymize the two, or at least highlight that synonymy may be reasonable.

3) The authors compiled a wonderful set of stratigraphic data, but do not really use it to draw any conclusions. Are any correlations available that might reveal something about the paleoecology of Anosteira, for instance a correlation with certain depositional systems, faunas, floras, or climate signals?

4) I find the discussion regarding plastral kinesis to be suboptimal as is:

a) The authors claim that kinesis is present, but do not state where. Indeed, the only sentence were this is discussed seems to grammatically imply that kinesis in the peripherals, which is doubtful. I therefore suggest spelling out to the reader where they believe kinesis to be found.

b) The presence of kinesis in carettochelyids is not really new, so announcing its presence for this taxon should be muted. Also, the authors extensively cite the work of Bramble in regards to plastral kinesis, but I cannot find any discussion of kinesis in carettochelyids in the cited papers. A quick literature search suggests instead that plastral kinesis had previously been noted for carettochelyids by Meylan (1988, Peltochelys), Meylan and Gaffney (1989, Adocus), or Hutchison et al. (2004, Burmemys), to name a few.

c) The authors state with doubt that posterior plastral kinesis is present in the females of Allaeochelys. Why is there any doubt?

d) Indeed, given the published record of sexual dimorphism in regards to posterior plastral kinesis, why don't the authors flag similar variation found in their sample?

A small number of additional points are highlighted in the attached PDF.

All in all, the requested changes are rather minor. I therefore highly recommend publication with modifications.

With best regards,

Walter Joyce

·

Basic reporting

This report reveals the relative abundance of Anosteira pulchra (Carettochelyidae) based on numerous unpublished materials during the middle Eocene in North America. Thus, this will offer a significant data on the taxonomic diversity of non-marine turtles during the Paleogene.

Experimental design

A comparative figure of A. pulchra and A, ornata would be useful for readers, if any.

Validity of the findings

This report seems important, as carettochelyids are relatively rare turtles in the fossil record in North America. This reveals the relative abundance of this family based on numerous unpublished materials during the middle Eocene in this area.

Additional comments

This report seems very important for understanding the morphological and stratigraphic diversities of hitherto little known A. pulchra. I hope this will be published very soon.

·

Basic reporting

This is an interesting and well-written manuscript. Several minor suggestions are indicated here (see General comments for the authors)

Experimental design

No comment

Validity of the findings

No comment

Additional comments

This is an interesting and well-written manuscript. Several minor suggestions are indicated here:
- Avoid using the abbreviation “NALMA” in the abstract, moving “North American Land Mammal Age (NALMA)” to the introduction.
- Some abbreviated generic names are used to start sentences both in the abstract and elsewhere in the manuscript (e.g. “…North America. A. pulchra is…”, “…taxa (Gilmore, 1916). B. inflata has…”). No generic name should be abbreviated at the beginning of a sentence.
- At the end of the introduction the authors indicate: “However, targeted collecting in recent years of Uintan herpetofauna in a measured stratigraphic framework has yielded 95 carettochelyid specimens”. I suggest completing this sentence indicating how many of these specimens were unpublished.
- Abbreviation section: Please divide the information into two subsections, each with a title (i.e., Institutional abbreviations; Anatomical abbreviations)
- Newly Referred Specimens subsection of the Systematic Paleontology section: add some more information in the text, at least referring to the number of individuals.
- Figures 3 and 4: I suggest not including specimen numbers in the figures: that information is indicated in the captions, and that should be enough.
- Figure 3: I think that the font size used for the abbreviations of the plates is very small.
- Caption of Fig. 6: Abbreviations should be removed from here, as they are indicated in the section Abbreviations.
- Figure 7: I suggest adding a scale to know the size of all the specimens, and not just of one of them. In addition, I believe that the photographs of some specimens of which a drawing is supplied have not been added in this work, but have not been published in previous papers either: Please add photograph of all specimens of which a drawing is provided.

I'm available for any additional clarification that authors and editors may need.

Cordially,

Adán

--
Dr. Adán Pérez-García

http://dfmf.uned.es/biologia/personal/aperez/

Grupo de Biología Evolutiva, Facultad de Ciencias, UNED
Madrid, Spain

---

## Round 0.2 · accepted · Accept

I am pleased to confirm that your paper has been accepted for publication.

·

Basic reporting

see below

Experimental design

see below

Validity of the findings

see below

Additional comments

I read the revised manuscript and the rebuttal latter and feel overall content with the implemented changes. I, therefore, suggest publication as is.
One final point: the layout team of PeerJ will likely squeeze the figures to only fill about two-thirds of the page to maintain the white margin of the left. If this is done, be sure to request that the figures are expanded to fill the full page. Thanks.
Walter Joyce